# *miR-124* controls male reproductive success in *Drosophila*

**Ruifen Weng[1,2], Jacqueline SR Chin[3], Joanne Y Yew[2,3], Natascha Bushati[3], Stephen M Cohen[1,2]\***

[1]Institute for Molecular and Cell Biology, Singapore, Singapore; [2]Department of Biological Sciences, National University of Singapore, Singapore, Singapore; [3]Temasek Life Sciences Laboratory, Singapore, Singapore

**Abstract** Many aspects of social behavior are controlled by sex-specific pheromones. Gender-appropriate production of the sexually dimorphic transcription factors doublesex and fruitless controls sexual differentiation and sexual behavior. *miR-124* mutant males exhibited increased male–male courtship and reduced reproductive success with females. Females showed a strong preference for wild-type males over *miR-124* mutant males when given a choice of mates. These effects were traced to aberrant pheromone production. We identified the sex-specific splicing factor *transformer* as a functionally significant target of *miR-124* in this context, suggesting a role for *miR-124* in the control of male sexual differentiation and behavior, by limiting inappropriate expression of the female form of *transformer*. *miR-124* is required to ensure fidelity of gender-appropriate pheromone production in males. Use of a microRNA provides a secondary means of controlling the cascade of sex-specific splicing events that controls sexual differentiation in *Drosophila*.

## Introduction

In animals, the performance of the individual in social behaviors such as mate recognition, courtship and aggression are important determinants of reproductive fitness. These behaviors are modulated in part by chemical cues, pheromones, used for intraspecific communication. In *Drosophila melanogaster*, the courtship and aggressive behaviors exhibited by male flies are influenced by a cocktail of pheromones produced by males and females (*Jallon, 1984*; *Fernandez et al., 2010*; *Wang and Anderson, 2010*). Detection of pheromones is mediated by specific receptors that detect compounds spread by volatile diffusion and transferred during physical contact (*Kurtovic et al., 2007*; *Vosshall, 2008*; *Stowers and Logan, 2010*; *Wang et al., 2011*; *Thistle et al., 2012*; *Toda et al., 2012*).

Pheromones in *Drosophila melanogaster* are strikingly sexually dimorphic in expression and in their effects on male and female behavior (*Jallon, 1984*; *Ferveur and Sureau, 1996*). Long-chained hydrocarbons present on the cuticular surface of the abdomen constitute a major class of *Drosophila* sex pheromones. The hydrocarbons are synthesized by specialized cells called oenocytes (*Billeter et al., 2009*). Female pheromones are largely comprised of *cis, cis*-7, 11-heptacosadiene and *cis, cis*-7, 11-nonacosadiene, both of which are known to serve as aphrodisiacs for males (*Antony et al., 1985*). Males primarily produce hydrocarbons bearing a single double bond (e.g., *cis*-7-tricosene, *cis*-7-pentacosene and *cis*-9-pentacosene), although these compounds are also produced by females (*Jallon and David, 1987*). The male-predominant *cis*-7-tricosene acts as an aphrodisiac for females but an anti-aphrodisiac for males. Members of the oenocyte-produced pentacosene family can also act as male aphrodisiacs, when present at high levels (*Scott and Richmond, 1988*; *Siwicki et al., 2005*).

*Drosophila* males also produce a different class of pheromones in the ejaculatory bulb, which are transferred during mating and mediate chemical communication (*Guiraudie-Capraz et al., 2007*;

**\*For correspondence:** scohen@ imcb.a-star.edu.sg

**Competing interests:** The authors declare that no competing interests exist.

**Reviewing editor**: Mani Ramaswami, Trinity College, Dublin, Ireland

**eLife digest** Like many animals, the fruit fly *Drosophila* uses pheromones to influence sexual behaviour, with males and females producing different versions of these chemicals. One of the pheromones produced by male flies, for example, is a chemical called 11-cis-vaccenyl-acetate (cVA), which is an aphrodisiac for female flies and an anti-aphrodisiac for males.

The production of the correct pheromones in each sex is genetically controlled using a process called splicing that allows a single gene to be expressed as two or more different proteins. A variety of proteins called splicing factors ensures that splicing results in the production of the correct pheromones for each sex. Sometimes, however, the process by which sex genes are expressed as proteins can be 'leaky', which results in the wrong proteins being produced for one or both sexes.

Small RNA molecules called microRNAs act in some genetic pathways to limit the leaky expression of genes, and a microRNA called *miR-124* carries out this function in the developing brain *Drosophila*. Now, Weng et al. show that *miR-124* also helps to regulate sex-specific splicing and thereby to control pheromone production and sexual behaviour.

Mutant male flies lacking *miR-124* were less successful than wild-type males at mating with female flies, and were almost always rejected if a female fly was given a choice between a mutant male and a wild-type male. Moreover, both wild-type and mutant male flies were more likely to initiate courtship behaviour towards another male if it lacked *miR-124* than if it did not.

The mutant male flies produced less cVA than wild-type males, but more of other pheromones called pentacosenes, which is consistent with the observed behaviour because cVA attracts females and repels males, whereas pentacosenes act as aphrodisiacs for male flies in large amounts. Weng et al. showed that these changes in the production of pheromones were caused by an increased expression of the female version of a splicing factor called *transformer* in the mutant males, but further work is needed to understand this process in detail.

*Yew et al., 2009*). 11-cis-Vaccenyl-Acetate (cVA), an oxygenated lipid, is thought to have an aphrodisiac effect on females, stimulating receptivity towards copulation, and acting as an anti-aphrodisiac for males (*Jallon, 1984*; *Cobb, 1996*; *Kurtovic et al., 2007*). CH503 (3-acetoxy-11,19-octacosadien-1-ol), a second lipid made in the male ejaculatory bulb, also acts as an anti-aphrodisiac for males after being transferred to the female during mating (*Yew et al., 2009*).

Sexually dimorphic behavior and chemical communication are under the control of the sex determination pathway (*Burtis and Baker, 1989*; *Ryner et al., 1996*; *Kimura et al., 2005*; *Villella et al., 2005*; *Vrontou et al., 2006*; *Kimura et al., 2008*; *Siwicki and Kravitz, 2009*). Expression of the splicing factor Sex-lethal (Sxl) in genetically female animals promotes sex specific splicing of the sexually dimorphic *transformer* transcript to produce the female splice form (Tra$^F$). Tra$^F$ in turn promotes splicing to produce the female form of Doublesex (Dsx$^F$). In the absence of Tra$^F$, the default male form of Dsx (Dsx$^M$) is produced, along with the male form of Fruitless (Fru$^M$). Dsx proteins direct male vs female sexual differentiation, including pheromone production, as well as sexual behavior (*Waterbury et al., 1999*; *Rideout et al., 2010*), whereas Fru$^M$ controls male sexual behavior but not pheromone production (*Demir and Dickson, 2005*; *Manoli et al., 2005*).

MicroRNAs have previously been implicated in the control of gene expression noise, acting as a backup mechanism to minimize the consequences of leaky expression of transcripts whose primary regulation is under transcriptional control (*Stark et al., 2005*; *Karres et al., 2007*; *Bushati et al., 2008*; *Shkumatava et al., 2009*; *Weng and Cohen, 2012*), reviewed in (*Herranz and Cohen, 2010*; *Ebert and Sharp, 2012*). miRNAs are also well suited to buffer the effects of inappropriate splicing. For example, *miR-1* can limit expression of the cytoplasmic splice form of tropomyosin, while sparing muscle specific splice forms (*Stark et al., 2005*). *miR-124* is abundantly expressed in the *Drosophila* brain, where it has been shown to limit leaky expression of an inhibitor of neuronal stem cell proliferation during larval development (*Weng and Cohen, 2012*). Here we present evidence that *miR-124* acts to limit the impact of leaky regulation of splicing in the sexual differentiation pathway. *miR-124* mutant males showed reduced mating success when paired with female flies, and elicited courtship by normal males. These effects were traced to aberrant pheromone production. We identified the sex-specific splicing factor *transformer* as the functionally significant target of *miR-124* in this process, suggesting

a role for *miR-124* in the control of male sexual differentiation, by limiting inappropriate expression of the female form of *transformer*.

## Results

### Aberrant male courtship behavior

*Drosophila* males engage in a complex set of courtship behaviors to induce receptiveness of females to mating. *miR-124* mutant males exhibited a normal repertoire of behaviors when paired with sexually mature Canton S (CS) female virgins in a standard courtship assay (including orientation toward the female, courtship song, tapping, licking, abdomen curling, and attempted copulation). However, *miR-124* mutant males achieved copulation significantly less often than CS controls during the 30-min observation period (**Figure 1A**, \*\*p<0.01). *miR-124* mutant females and CS females did not show a significant difference in receptiveness to courtship by CS males (**Figure 1—figure supplement 1**).

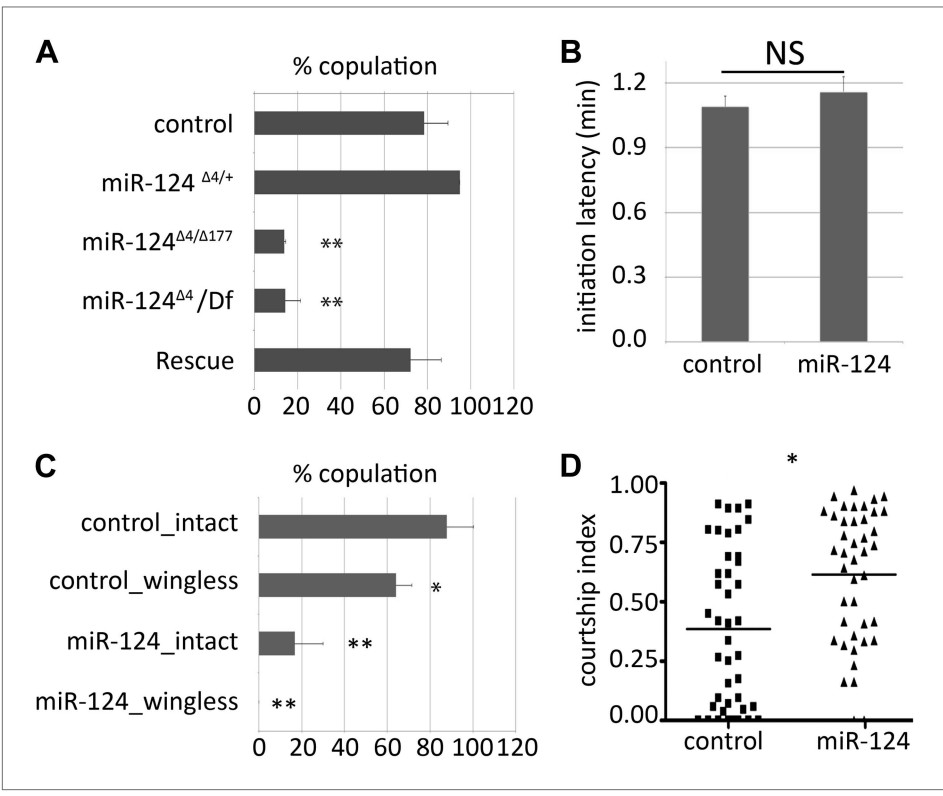

**Figure 1**. Male courtship behavior. (**A**) Percentage of males achieving copulation in a 30-min observation period. Genotypes as indicated. Control males were CS. Rescue indicates the *miR-124* RMCE allele with *miR-124* reintegrated at the endogenous locus (34). Data represent the average of five independent experiments ± SEM. (**B**) Courtship initiation latency measures time (in minutes) to initiate courtship for CS control and *miR-124* flies. Data represent the average of four independent experiments ± SEM. ns: no significant difference. (**C**) Percentage of males achieving copulation in 30 min, comparing CS control and *miR-124* mutant flies before and after removal of the wings. Data represent the mean of more than 20 movies per genotype ± SD. (**D**) Courtship index compares the proportion of the measurement period males spent courting. CS control and *miR-124* mutant males were tested using decapitated CS females as targets. Data represent 56 trials conducted in 4 batches of 14 pairs each. The horizontal line represents the median. Although the variance was high, the difference in the medians was borderline significant (p=0.042 comparing for the 56 pairs). However, when the data were analyzed as the average of four independent experiments (n = 14 in each experiment) the difference in the means was not significant. In all figures: \*p<0.05, \*\*p<0.01, \*\*\*p<0.001, ns: not statistically significant.

The following figure supplements are available for figure 1:

**Figure supplement 1**. Receptivity of *miR-124* mutant females.

To determine the basis for the reduced mating efficiency we examined a number of courtship behavioral parameters. Initiation latency, the time taken by the male to recognize the female and begin courtship, was unaffected (**Figure 1B**). Males use a courtship song produced by wing vibration to elicit receptivity in female flies. If a defect in courtship song is responsible for the poor mating success of *mir-124* mutant males, removal of the wings should eliminate the observed difference in receptivity of females to courtship. Under these conditions, *miR-124* mutants were also less successful in mating than control males (**Figure 1C**). Thus, courtship song does not appear to be an important contributor to the difference between control and mutant males.

Progression from courtship to copulation involves behavioral input from female flies (**Rezaval et al., 2012**). To remove female behavioral response from the assay, we tested decapitated target flies. We did not observe a reduction in the level of courtship activity by mutant males compared to that of control males under these conditions (**Figure 1D**). Thus the failure to achieve copulation is unlikely due to reduced activity of the mutant male. Reduced copulation therefore likely reflects rejection of the *miR-124* mutant male's advances by the female. This defect was rescued when *miR-124* expression was restored in the miRNA expressing cells of the mutant (**Figure 1A**).

### *miR-124* mutant males induce aberrant behavior in other males

*Drosophila* males normally pay little sexual attention to other sexually mature males. Males with altered sexual orientation elicit a behavior called chaining, in which groups of males follow each other while attempting courtship (**Hall, 1978**). We observed chaining among groups of *miR-124* mutant males. Male–male courtship could result from altered sexual orientation or from a change in the expression of inhibitory or stimulatory cues, or from an inability to recognize inhibitory courtship cues. To distinguish among these possibilities, we quantified the courtship behavior of mutant and control males when placed with mutant or control male targets. There was no difference in the amount of time that *miR-124* mutant or CS control males devoted to courtship of CS target males (**Figure 2A**). However, *miR-124* mutant targets elicited more courtship activity from both CS control and *miR-124* males (**Figure 2A**, **p<0.01). This effect was suppressed when *miR-124* expression was restored in its endogenous domain (**Figure 2B**, **p<0.01). Next, a courtship choice assay was performed in which test males were presented with a choice of decapitated control or *miR-124* target males. Wild-type CS males devoted more than twice as much time to courting the *miR-124* target as they did to the control target (**Figure 2C**, **p<0.01). Thus, CS males appeared to be more attracted by *miR-124* males than by other CS males.

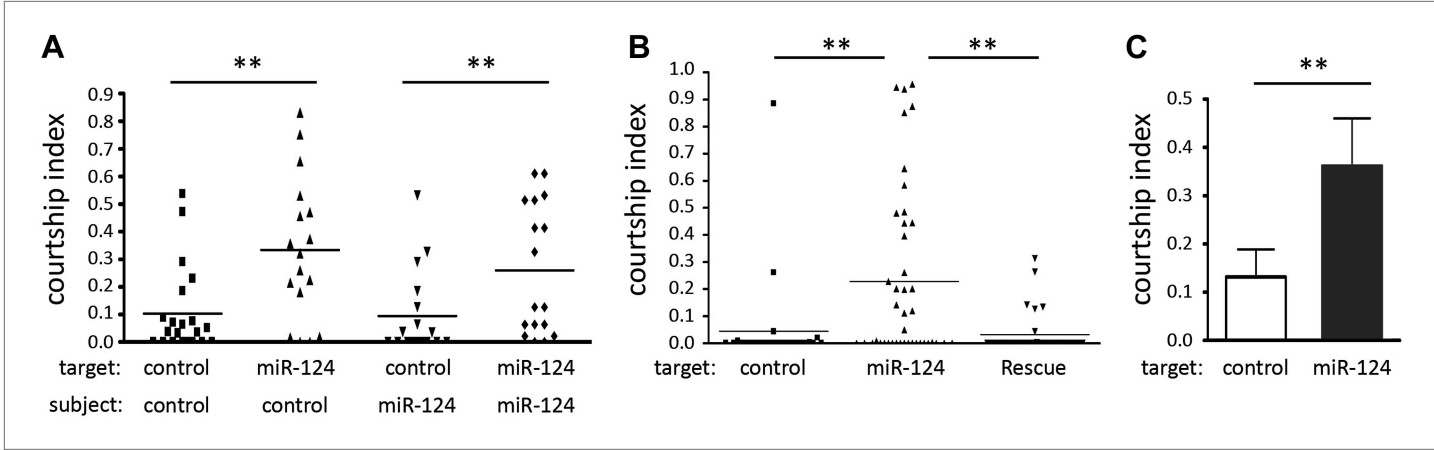

**Figure 2**. Male–male courtship. (**A**) Courtship index comparing CS control and *miR-124* mutant flies using decapitated CS or *miR-124* mutant males as targets. The number of animals used for each sample is indicated (n:). Scores for many control flies were very close to zero, overlapping the X axis, and so are not visible as individual data points in the scatter plot. Data represent one of four independent trials performed with comparable results. (**B**) Courtship index for CS control males toward decapitated targets. The target genotypes used are CS control, *miR-124* mutant and rescued mutant. Data represent one of four independent trials performed with comparable results. (**C**) Courtship choice assay comparing the time CS control males courted decapitated CS control and *miR-124* mutant targets, when presented together. Data represent the mean of more than 20 movies per genotype ± SD.

The behavior of the control and mutant males in each of these assays depended on the genotype of the targets, not on the genotypes of the test males themselves. This suggests that the male–male courtship phenotype is unlikely to reflect a change in neuronal circuitry of the mutant males that could affect their sexual orientation or their ability to recognize normal inhibitory cues. Rather, the observation that behaviorally inert mutant males elicited courtship behavior from control males suggested a change in chemical cues provided by *miR-124* mutant males.

## Aberrant pheromone production by *miR-124* mutant males

Cuticular hydrocarbon profiles were generated for sexually mature *miR-124* mutant and control male flies using gas chromatography/mass spectrometry (GC-MS). GC-MS analysis showed that the level of cVA was significantly reduced in *miR-124* mutant males (*Table 1* and *Figure 3A*, ***p<0.001), and was partially restored in rescued mutants (*Table 1* and *Figure 3A*). Conversely, pentacosenes were recovered at elevated levels on *miR-124* mutant males by GC-MS (*Table 1* and *Figure 3B*, *p<0.05) and found near normal levels in the rescue mutants (*Table 1* and *Figure 3B*, **p<0.01). These results suggest that *miR-124* mutant males produce elevated levels of compounds that behave as male aphrodisiacs, and lower levels of compounds that have anti-aphrodisiac activity on males, leading to increased male–male courtship.

**Table 1.** GC-MS analysis of cuticular hydrocarbon extracts from control, *mir-124* mutant, and rescued mutant males

| Compound and elemental composition* | Control† (n = 6) | *mir-124* mutant† (n = 6) | Rescued mutant† (n = 6) |
|---|---|---|---|
| C21:0 (nC21) | 0.46 ± 0.08 | 0.32 ± 0.05 | 0.76 ± 0.11 |
| C22:1 | 0.24 ± 0.01 | 0.27 ± 0.02 | 0.35 ± 0.02 |
| cVA (cis-vaccenyl acetate) | 9.36 ± 3.40 | 1.75 ± 0.57*** | 6.60 ± 2.17*** |
| C22:0 | 0.74 ± 0.06 | 0.62 ± 0.02 | 0.95 ± 0.05 |
| 7,11-C23:2 | 0.13 ± 0.01 | 0.07 ± 0.001 | 0.12 ± 0.02 |
| 9-C23:1 (9-tricosene) | 1.39 ± 0.13 | 1.76 ± 0.25 | 1.84 ± 0.14 |
| 7-C23:1 (7-tricosene) | 23.52 ± 1.17 | 24.92 ± 1.74 | 32.80 ± 2.03*** |
| 5-C23:1 (5-tricosene) | 2.71 ± 0.11 | 3.06 ± 0.20 | 3.01 ± 0.18 |
| C23:0 (nC23) | 10.57 ± 0.40 | 11.21 ± 0.25 | 12.66 ± 0.63** |
| C24:1 | 0.32 ± 0.11 | 0.37 ± 0.09 | 0.30 ± 0.07 |
| C24:0 | 0.36 ± 0.02 | 0.43 ± 0.04 | 0.35 ± 0.03 |
| 2-MeC24 | 1.44 ± 0.08 | 1.58 ± 0.15 | 2.03 ± 0.12 |
| C25:2 | 0.52 ± 0.06 | 0.71 ± 0.07 | 0.70 ± 0.04 |
| 9-C25:1 (9-pentacosene) | 4.80 ± 0.61 | 6.33 ± 0.65* | 4.11 ± 0.74 |
| 7-C25:1 (7-pentacosene) | 22.99 ± 1.55 | 25.62 ± 0.63*** | 11.61 ± 1.16*** |
| 5-C25:1 (5-pentacosene) | 1.10 ± 0.33 | 0.79 ± 0.02 | 2.38 ± 0.01 |
| C25:0 (nc25) | 2.34 ± 0.15 | 3.13 ± 0.03 | 2.52 ± 0.15 |
| 2-MeC26 | 6.75 ± 0.49 | 5.37 ± 0.08 | 6.55 ± 0.13 |
| 9-C27:1 | 0.16 ± 0.02 | 0.19 ± 0.03 | 0.12 ± 0.04 |
| 7-C27:1 | 0.97 ± 0.10 | 0.77 ± 0.07 | 0.29 ± 0.06** |
| C27:0 (nC27) | 1.66 ± 0.33 | 2.42 ± 0.60 | 1.86 ± 0.39 |
| 2-MeC28 | 5.90 ± 0.81 | 5.95 ± 0.71 | 6.18 ± 0.77 |
| C29:0 | 0.37 ± 0.11 | 0.78 ± 0.26 | 0.54 ± 0.17 |
| 2-MeC30 | 0.64 ± 0.16 | 0.99 ± 0.27 | 0.87 ± 0.25 |

*The elemental composition is listed as the carbon chain length followed by the number of double bonds; 2-Me indicates the position of methyl branched compounds.
†The normalized signal intensity for each compound and SEM is indicated; *p<0.05, **p<0.01, ***p<0.001 when compared to control (ANOVA followed by post-hoc Tukey HSD test).

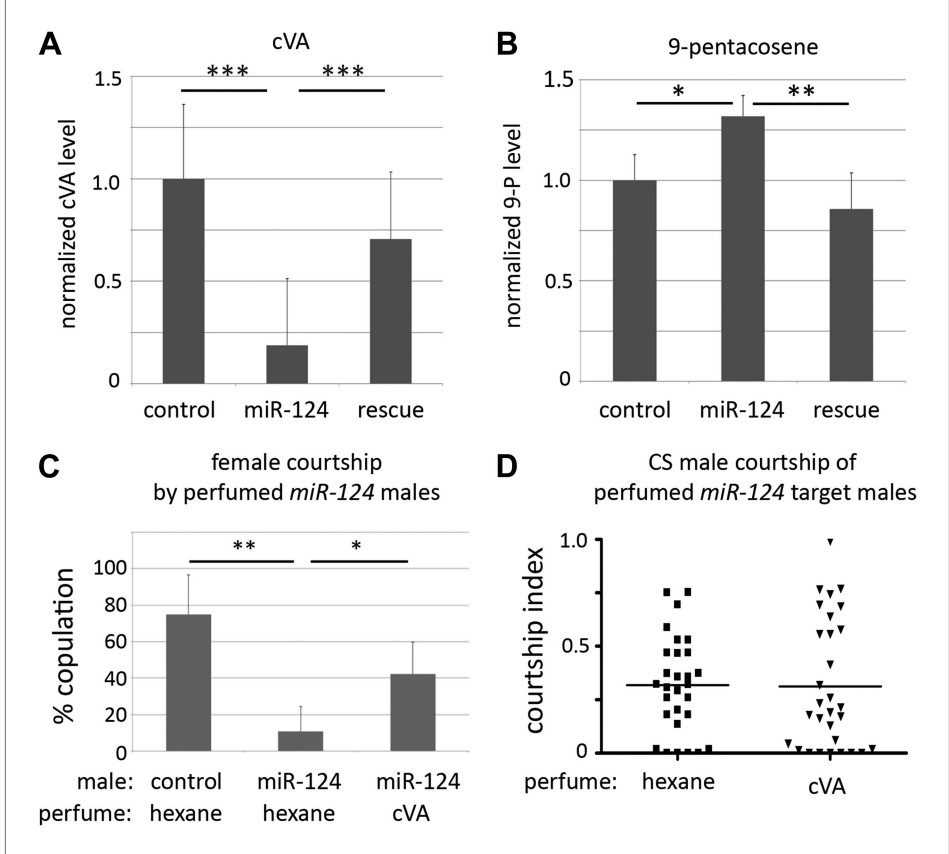

**Figure 3**. Aberrant pheromone production by *miR-124* mutant males. (**A**) Normalized cVA level measured by GC-MS in extracts from control, *miR-124* mutant, and rescued mutant males. Data represent the average of six independent preparations ± SEM. n = 15 in each preparation. (**B**) Normalized 9-pentacosene level measured by GC-MS from control, *miR-124* mutant, and rescued mutant males. Data represent the average of six independent preparations ± SEM. n = 15 in each preparation. (**C**) Percentage of males achieving copulation in 30 min, comparing *miR-124* mutant flies with or without cVA perfuming. Hexane perfuming was used as a control. Data represent the mean of >20 movies per genotype ± SD. (**D**) Courtship index (CI) using CS test males and *miR-124* mutant target males perfumed with hexane solvent alone as a control, or with hexane containing cVA. No significant difference was observed between CI of CS males towards *miR-124* target males perfumed with hexane (average CI = 0.320) or with cVA (average CI = 0.315). n = 30 in each experiment.

The following figure supplements are available for figure 3:

**Figure supplement 1**. Abundance of cVA on perfumed flies.

To ask whether the changes in pheromone levels were sufficient to account for the increased male–male courtship elicited by *miR-124* mutant males, we carried out perfuming experiments. Mutant males perfumed with cVA showed a significant improvement in their ability to achieve copulation with control females (***Figure 3C***, *p<0.05). We also examined the effects of perfuming on male courtship behavior. Decapitated *miR-124* mutant males were perfumed with cVA and used as targets in the male–male courtship assay. There was no significant difference between courtship of targets perfumed with cVA or with the hexane solvent alone (***Figure 3D***; the perfuming protocol restored cVA to <50% the level on control flies, ***Figure 3—figure supplement 1***). The cVA-perfumed *miR-124* mutant target males also have elevated levels of the pentacosene pheromones. Thus, the perfumed mutant males are expected to give mixed excitatory and inhibitory courtship signals. In this context, the level of cVA reached by perfuming may be insufficient to fully rescue male–male courtship, while being sufficient to restore male–female courtship. However, we do not exclude the possibility that cVA might be more effective at inhibiting male courtship if presented at a higher local concentration. cVA is normally

concentrated on the tip of the male ejaculatory apparatus. The perfuming experiment distributes cVA over the entire body.

## Consequences of aberrant pheromone production

Although *miR-124* mutant males showed less mating success in the courtship assay, they are fertile in laboratory conditions. The aberrant pheromone production might be expected to confer a disadvantage in a competitive situation, where the female has a choice of mates. To test this, single CS female virgins were placed in mating chambers with one CS control male and one *miR-124* mutant or rescued mutant male. *miR-124* mutant males were rarely selected in the presence of a wild-type male, but females did not distinguish between CS males and rescued mutant males (***Figure 4A***). Mutant males would likely be at a disadvantage in a natural competitive setting.

Aggression is another social behavior commonly observed among *Drosophila* males, and is promoted by chemical cues such as cVA (***Wang and Anderson, 2010***). To ask if the loss of *miR-124* influences male aggressiveness, the fighting behavior between pairs of mutant or wild-type males was analyzed. In this setting, wild-type males typically fight for sole occupancy of the food patch, resulting in the establishment of a hierarchy (***Chen et al., 2002***). *miR-124* mutant males exhibited overall lower levels of aggression based on several parameters. First, mutant males experienced more encounters before any fighting took place (latency, ***Figure 4B***, \*\*p<0.01). Mutant males exhibited lower frequency of fighting behaviors, including lunging and fencing (***Figure 4C***, \*\*p<0.01) and were often observed sharing the food patch after a few encounters. There was no obvious difference in overall activity levels, based on observation during the assay and results of a locomotion assay (***Figure 4—figure supplement 1***). Lower cVA production in the *miR-124* mutant may contribute to the lowered intensity of aggressive behaviors observed in these flies.

## *miR-124* acts in the sex determination pathway in the CNS

Sexually dimorphic behavior and chemical communication are under the control of the sex determination pathway (***Figure 5A***). To ask whether *miR-124* might act in the sex determination pathway, we used a microRNA sponge to deplete *miR-124* in *doublesex*-expressing cells. *Doublesex* expression is sexually dimorphic in the brains of males and females (***Rideout et al., 2010***; ***Robinett et al., 2010***). In the male, Dsx$^M$ is required for differentiation of Fru$^M$-expressing neurons (***Rideout et al., 2010***). To

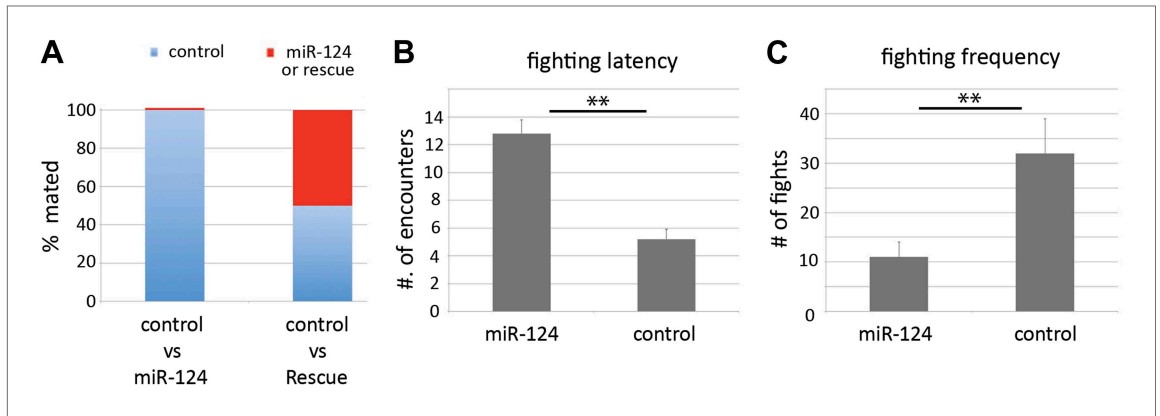

**Figure 4**. Comparison of other social behaviors. (**A**) Female mate choice was monitored by videotaping in chambers containing single females and two males of the indicated genotypes. The genotype of the male that succeeded in copulating was recorded. More than 95% of control male achieved copulation, in the presence of *miR-124* mutant males (left bar) compared with ~50% in the presence of rescued mutant males (right bar). (**B**) Fighting latency was monitored by videotaping encounters between pairs of males in chambers containing a patch of food. Latency is the number of encounters that do not elicit aggressive behavior prior to the first fight. Data represent the mean of more than 16 movies per genotype ± SD. (**C**) Fighting frequency was monitored by videotaping encounters between pairs of males in chambers containing a patch of food. Frequency records the number of aggressive encounters in 30 min. Data represent the mean of more than 16 movies per genotype ± SD.

The following figure supplements are available for figure 4:

**Figure supplement 1**. Locomotion assay.

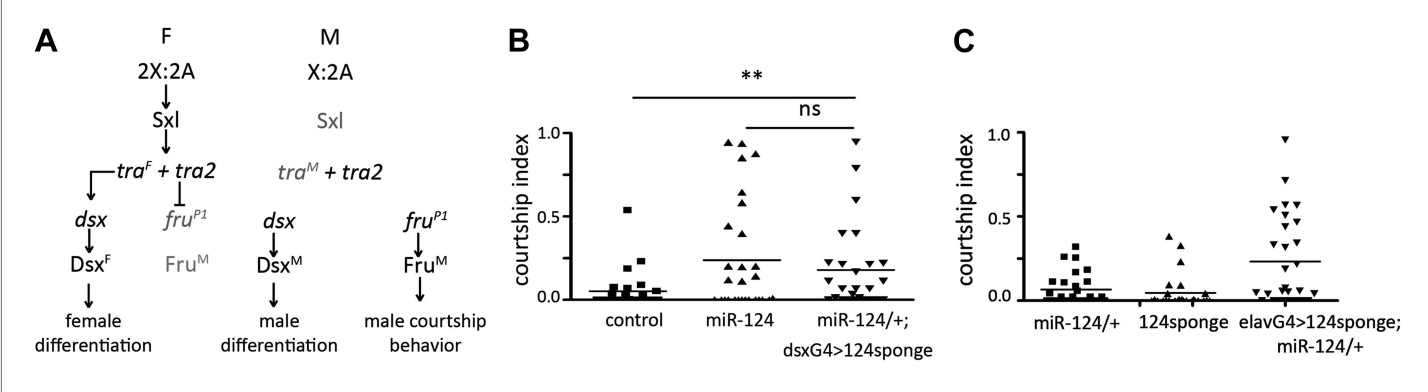

**Figure 5**. *miR-124* acts in the sex differentiation pathway. (**A**) Key components of the sexual differentiation system. (**B**) Courtship index comparing *miR-124* mutants and flies expressing a *miR-124* sponge under *dsx-Gal4* control in males lacking one copy of the endogenous *miR-124* gene with CS controls. n: number of animals per sample. Data represent one of four independent trials performed, with comparable results. The horizontal lines represent the median for each data set. (**C**) Courtship index comparing proportion of time CS control males spent courting decapitated male flies that had expressed the *UAS-miR-124* sponge under *elav-Gal4* control vs flies that carried the *UAS-miR-124* sponge transgene without Gal4 and vs *miR-124* heterozygous control males. n: number of animals per sample. Data represent one of four independent trials performed, with comparable results.

increase efficacy, the sponge was expressed in males lacking one copy of the endogenous *miR-124* gene. Depletion of *miR-124* in *dsx*-expressing cells elicited male–male courtship at a level comparable to that elicited by homozygous *miR-124* null mutant target males (***Figure 5B***).

Doublesex is expressed in both neuronal and non-neuronal tissues, whereas *miR-124* is highly enriched in the CNS. To ask whether the CNS is the site of *miR-124* action, we used the pan-neuronal *elav-Gal4* driver to direct expression of the *miR-124* sponge in males lacking one copy of the endogenous *miR-124* gene. This resulted in increased courtship of these flies by wild-type males (***Figure 5C***), suggesting that *miR-124* activity in the CNS contributes to the male courtship phenotype, presumably by modulation of pheromone production.

Computational target prediction datasets do not list any of the known components of the sex determination pathway among predicted *miR-124* targets. To allow for the possibility that the prediction algorithms might miss sites with specific features, we scanned sex determination pathway transcripts using the RNAhybrid prediction tool (***Rehmsmeier et al., 2004***) and found two potential sites for *miR-124* in the 3′ UTR of *transformer* (***Figure 6A,B***). The first site is present in the 3′ UTR region common to both the female-specific and non-sex-specific *tra* transcripts, while the second one is located in sequences unique to the non-sex-specific form. Pairing to residues 2–8 of the miRNA, called the seed region, is important in miRNA target identification (***Brennecke et al., 2005***). Each of the sites in *tra* would require 3 G:U base pairs with the *miR-124* seed. G:U base pairs in the seed region are compatible with miRNA function, but reduce the efficiency of target regulation (***Brennecke et al., 2005***). A luciferase reporter assay showed that these sites can mediate regulation by *miR-124* (***Figure 6C***).

As a first step to determine whether *tra* might be a functionally important target of *miR-124* in vivo, we examined *tra* transcript levels by quantitative RT-PCR in RNA samples from control and *miR-124* mutant male heads. The *tra* primary transcript undergoes sex-specific splicing in females to produce *tra^F*, which encodes a splicing factor (***Figure 6B***). An alternate splice form is produced in both males and females, and is thought to produce a non-functional protein. Using primers that recognize the female-specific form, we observed that *tra^F* mRNA increased ~2.5-fold in the mutant and returned to near normal levels in the rescued mutant (***Figure 7A***, *p<0.05). The female-specific *tra^F* splice form can be detected at low levels in control males by qPCR, at a few percent of the level found in females (***Figure 7—figure supplement 1***). This likely reflects a low level of improper splicing.

Consistent with previous reports (***Chan and Kravitz, 2007***; ***Fernandez et al., 2010***), increased expression of TraF in the male brain proved to be sufficient to reduce mating success and to elicit male–male courtship (not shown). If elevated *tra^F* expression contributes to the *miR-124* mutant phenotype, we would expect reducing *tra^F* levels to ameliorate the mutant phenotype. For these experiments, a *UAS-tra^RNAi* transgene was expressed under *miR-124-Gal4* control in the *miR-124* mutant

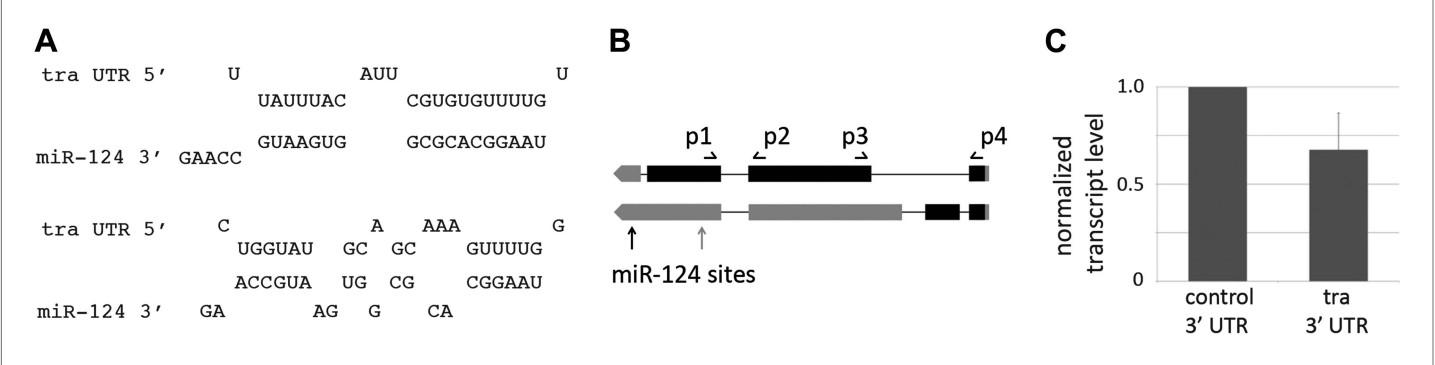

**Figure 6**. *miR-124* targets *transformer*. (**A**) Predicted pairing of *miR-124* to two sites in the *tra^F* transcript. (**B**) Sex-specific splicing results in the formation of a female-specific *tra^F* isoform. A non-sex-specific isoform is produced in males and females, *tra^C*. Exons are represented by black boxes, 5′ UTR and 3′ UTR by grey boxes. Sites for the primer-pairs used for detection of both isoforms, p1 and p2, span an intron in both splice forms. The PCR product from the spliced mRNA is 87 bp (unspliced primary transcript would produce a product of 154 bp). Primers p3 and p4 span the first intron of *tra^F*. Note that the positions of the primer pairs are approximate. The positions of the 2 *miR-124* target sites are indicated. (**C**) Luciferase reporter assays. S2 cells were transfected to express a *tra* 3′UTR luciferase reporter or a control reporter with the SV40 3′UTR. Cells were co-transfected to express *miR-124* or with a vector-only control, and a Renilla luciferase reporter as a control for transfection efficiency. Data show the mean ratio of firefly to Renilla luciferase activity based on three independent replicates. Error bars represent SEM. $p < 0.05$ using two-tailed unpaired Student's *t*-test.

background. The transgene targets a region common to both the female and non-sex-specific splice forms. Lowering *tra^F* levels in the *miR-124* expressing cells was sufficient to increase male–female mating success (*Figure 7B*); to reduce male–male courtship (*Figure 7C*), to improve production of cVA by several fold (*Table 2* and *Figure 7D*), and to lower levels of 9-pentacosene (*Table 2* and *Figure 7E*). Lowering *tra^F* levels in neurons by expressing *UAS-tra^RNAi* under *elav-Gal4* control also proved to be sufficient to suppress male–male courtship (*Figure 7F*). These findings indicate that upregulation of *transformer* in the CNS of the *miR-124* mutant is causally linked to the pheromone production and behavioral abnormalities in the mutant males.

## Discussion

### *miR-124* suppresses the consequences of leaky splicing

It is generally thought that the sex determination pathway acts in a binary fashion, with particular spliced forms of the pathway being turned on or off, depending on the genetic sex of the cell (*Cline and Meyer, 1996*). The Sxl splicing factor is produced in genetic females and competes with U2AF, an essential splicing factor, for binding to a splice site in the *tra* primary transcript. In the presence of sufficient Sxl, U2AF binds to a lower affinity site and promotes splicing to produce the female-specific *tra^F* transcript (*Valcarcel et al., 1993*). Nonetheless, low-levels of the female-specific *Sxl* and *tra^F* transcripts have been observed in males (this report; *Tarone et al., 2005*). In the case of *tra^F*, this might reflect a low-level of U2AF binding to the low affinity site, even in the absence of Sxl. Leaky low-level expression of Sxl in males could be another contributing factor. Under normal conditions, the level of *traF* transcript found in males appears to be innocuous.

Inappropriate splicing to produce *tra^F* transcript in males is expected to increase production of *dsx^F* at the expense of *dsx^M*. Interestingly, the modest increase in the level of *tra^F* in *miR-124* mutant males led to reduced splicing of *dsx* to produce *dsx^M*, but we did not observe a corresponding increase in the production of the female splice form *dsx^F* (*Figure 8*). Production of *dsx^F* requires the assembly of a complex containing Tra^F protein along with Tra2 and SR proteins at a series of sites that comprise the female-specific splice enhancer (*Lynch and Maniatis, 1996*). Our findings suggest that a modest increase in the level of Tra^F protein can interfere with production of *dsx^M* without leading to production of *dsx^F*. If low levels of Tra^F protein can lead to assembly of non-functional complexes, it is possible that their binding to the female-specific splice enhancer, might compromise male splicing without effectively promoting female splicing.

When expressed at high levels, *tra^F* or *dsx^F* can compromise male sexual differentiation and behavior (*Mckeown et al., 1988*; *Villella and Hall, 1996*). Our findings provide evidence that a modest

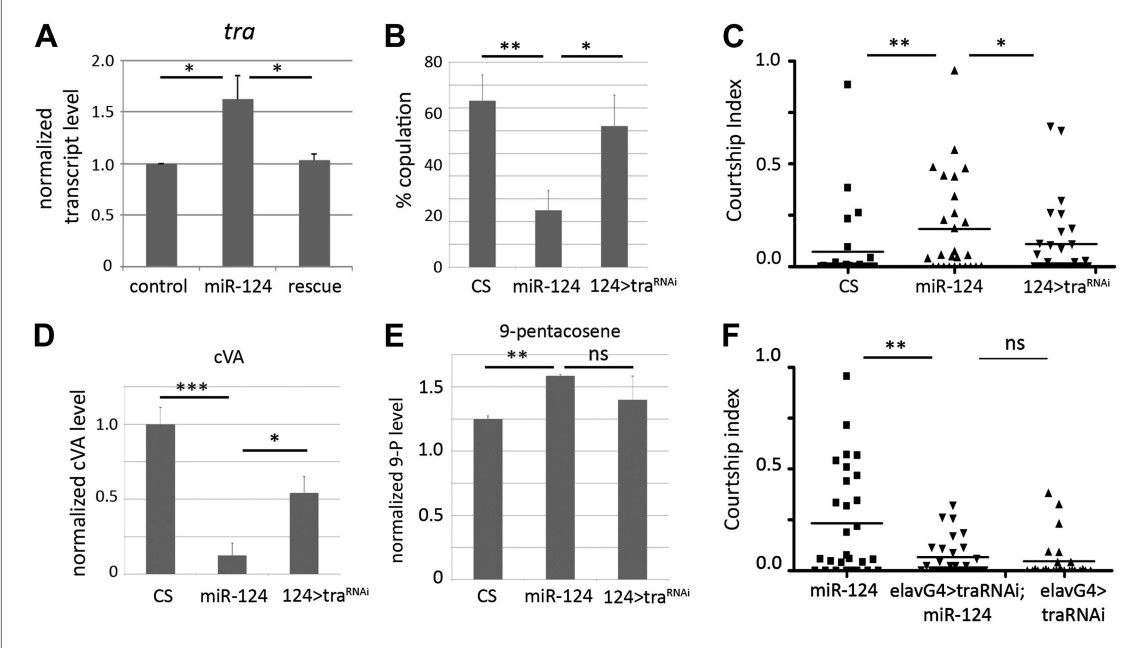

**Figure 7**. *miR-124* acts through regulation of *transformer*. (**A**) Elevated expression of *tra*[F] transcript measured by quantitative real-time PCR using RNA isolated from male flies of the indicated genotypes (primer pair p3 and p4). *actin 42A* was used as an internal control for normalization. Data represent the average of five independent experiments ± SEM. Although tra[F] transcript levels are low in control males, they were detected by quantitative real-time PCR (traces are shown in **Figure 6**). (**B**) Percentage of males achieving copulation with CS females in 30 min. Data represent the mean of more than 20 movies per genotype ± SD. Genotypes: CS: canton S control; *miR-124* refers to the trans-heterozygous mutant combination *miR-124[Δ4]/miR-124 [Δ177]*; 124>tra[RNAi] refers to the trans-heterozygous mutant combination *miR-124[Δ4]/miR-124 [Δ177]* carrying the *miR-124*-promoter Gal4 transgene and UAS-tra [RNAi]. Depletion of *tra* significantly improved performance of the *miR-124* mutant males. (**C**) Courtship index comparing proportion of time CS control males spent courting decapitated males of the indicated genotypes. n: number of animals per sample. Data represent one of three independent trials performed, with comparable results. Depletion of *tra* significantly reduced the attractiveness of the *miR-124* mutant males to normal levels. (**D**) Quantification of cVA levels in males of the indicated genotypes by GC-MS. Knocking down of *tra* in using miR-124Gal4 driver significantly rescued the changes cVA levels in *miR-124* mutant males. Data represent the average of two (for miR-124 > tra[RNAi]) or three replicates (CS and *miR-124*) ±SEM. n = 15 in each replicate. (**E**) Quantification of 9-pentacosene levels in males of the indicated genotypes by GC-MS. Depletion of *tra* lowered 9-pentacosene levels to within control levels. Data represent the average of two (for miR-124 > tra[RNAi]) or three replicates (CS and *miR-124*) ±SEM. n = 15 in each replicate. (**F**) Courtship index comparing proportion of time CS control males spent courting decapitated males of the indicated genotypes. N = 28 animals per sample. Data represent one of three independent trials performed, with comparable results.

The following figure supplements are available for figure 7:

**Figure supplement 1**. Amplification of *tra*[F] shown by quantitative real-time RT-PCR.

increase in the level of *tra*[F] in *miR-124* expressing cells in the CNS can interfere with male pheromone production. In this scenario microRNA mediated regulation ensures that leakiness in the production of *tra*[F] is kept at levels that are functionally insignificant in the male. A modest increase in *tra*[F] is not expected to have much effect in females, where the endogenous level is higher. microRNAs are well suited to provide an additional layer of noise reduction to post-transcriptional regulation mediated by splicing.

## *miR-124* is required for proper male-specific pheromone production

Pheromone production is controlled by the sex determination pathway. Genetic experiments have demonstrated the role of the Dsx protein in the regulation of male and female specific pheromone profiles. In females, Dsx[F] ensures the production of female-specific hydrocarbons while suppressing the production of male-specific hydrocarbons and other male-specific pheromones such as cVA. The presence of Dsx[M] protein in males ensures that synthesis of female-specific hydrocarbons are suppressed in males (***Baker and Belote, 1983***; ***Waterbury et al., 1999***).

In animals lacking *miR-124*, the level of *tra* transcripts increases. The presence of Tra[F] is expected to affect sexual differentiation in males. Gal4-directed expression of Dsx[F] in an otherwise wild-type

**Table 2.** GC-MS analysis of cuticular hydrocarbon extracts from control, *miR-124* mutant, rescued mutants, and *miR-124>tra-RNAi* males

| Compound and elemental composition* | Control† (n = 3) | *mir-124* mutant† (n = 3) | Rescued mutant† (n = 3) | mir-124> tra-RNAi† (n = 2) |
|---|---|---|---|---|
| C21:0 (nC21) | 0.28 ± 0.1 | 0.21 ± 0.01 | 0.51 ± 0.03 | 0.35 ± 0.04 |
| C22:1 | 0.22 ± 0.02 | 0.24 ± 0.01 | 0.31 ± 0.02 | 0.34 ± 0.03 |
| cVA (cis-vaccenyl acetate) | 3.86 ± 0.43 | 0.48 ± 0.04*** | 2.57 ± 0.47* | 2.09 ± 0.23* |
| C22:0 | 0.61 ± 0.03 | 0.60 ± 0.01 | 0.87 ± 0.05 | 0.70 ± 0.05 |
| 7,11-C23:2 | 0.14 ± 0.01 | 0.07 ± 0.001 | 0.17 ± 0.02 | 0.11 ± 0.02 |
| 9-C23:1 (9-tricosene) | 1.10 ± 0.05 | 1.20 ± 0.02 | 1.57 ± 0.12 | 1.94 ± 0.07 |
| 7-C23:1 (7-tricosene) | 21.68 ± 1.14 | 21.04 ± 0.29 | 29.07 ± 2.12*** | 28.95 ± 2.20*** |
| 5-C23:1 (5-tricosene) | 2.56 ± 0.05 | 2.62 ± 0.05 | 2.71 ± 0.25 | 3.11 ± 0.40 |
| C23:0 (nC23) | 9.84 ± 0.15 | 10.66 ± 0.06 | 11.33 ± 0.2* | 10.35 ± 0.33 |
| C24:1 | 0.09 ± 0.05 | 0.19 ± 0.01 | 0.16 ± 0.02 | 0.22 ± 0.01 |
| C24:0 | 0.41 ± 0.01 | 0.52 ± 0.01 | 0.40 ± 0.03 | 0.44 ± 0.01 |
| 2-MeC24 | 1.52 ± 0.13 | 1.24 ± 0.02 | 1.81 ± 0.15 | 1.78 ± 0.09 |
| C25:2 | 0.41 ± 0.02 | 0.54 ± 0.02 | 0.74 ± 0.01 | 0.76 ± 0.06 |
| 9-C25:1 (9-pentacosene) | 6.13 ± 0.12 | 7.78 ± 0.05** | 5.74 ± 0.21 | 6.86 ± 1.02 |
| 7-C25:1 (7-pentacosene) | 26.01 ± 0.69 | 26.97 ± 0.25 | 14.09 ± 0.46*** | 23.23 ± 1.15*** |
| 5-C25:1 (5-pentacosene) | 1.41 ± 0.68 | 0.75 ± 0.01 | 023 ± 0.02 | 0.59 ± 0.03 |
| C25:0 (nc25) | 2.65 ± 0.06 | 3.79±0.05 | 2.85 ± 0.07 | 2.68 ± 0.26 |
| 2-MeC26 | 7.72 ± 0.28 | 5.22 ± 0.01*** | 6.64 ± 0.27 | 5.23 ± 0.21*** |
| 9-C27:1 | 0.20 ± 0.01 | 0.25 ± 0.01 | 0.20 ± 0.02 | 0.18 ± 0.05 |
| 7-C27:1 | 1.15 ± 0.09 | 0.92 ± 0.02 | 0.41 ± 0.01 | 0.60 ± 0.08 |
| C27:0 (nC27) | 2.38 ± 0.12 | 3.77 ± 0.08* | 2.72 ± 0.16 | 1.92 ± 0.21 |
| 2-MeC28 | 7.66 ± 0.13 | 7.53 ± 0.05 | 7.85 ± 0.37 | 5.76 ± 0.33** |
| C29:0 | 0.62 ± 0.06 | 1.37 ± 0.01 | 0.92 ± 0.1 | 0.52 ± 0.05 |
| 2-MeC30 | 0.98 ± 0.06 | 1.59 ± 0.03 | 1.41 ± 0.13 | 0.66 ± 0.02 |

*The elemental composition is listed as the carbon chain length followed by the number of double bonds; 2-Me indicates the position of methyl branched compounds.
†The normalized signal intensity for each compound and SEM is indicated; *p<0.05, **p<0.01, ***p<0.001 when compared to control (ANOVA followed by post-hoc Tukey HSD test).

male (also expressing DsxM) has been reported to reduce cVA levels, whereas DsxF expression in *dsx* mutant males abolished cVA production completely (***Waterbury et al., 1999***).

Ectopic expression of DsxF in XY males has also been shown to cause production in female-specific diene-hydrocarbons such as *cis, cis-7*, 11-heptacosadiene and *cis, cis-7*, 11-nonacosadiene (***Waterbury et al., 1999***). We did not detect these compounds in cuticular extracts from *miR-124* mutant males. The difference is likely due to the absence of *miR-124* expression in the oenocytes where the TraF–DsxF cascade is thought to exert its effect on female hydrocarbon production.

Regulation of male-specific hydrocarbons is probably more complex and is likely to involve modulation from the nervous system. Many of the characteristic male compounds are also synthesized by the oenocytes, since genetic ablation of these cells abolished all male hydrocarbon production, but does not affect levels of cVA, produced in the ejaculatory bulb (***Billeter et al., 2009***). However, feminization of the nervous system in XY males led to significant elevation of characteristic male hydrocarbons such as cis-7-tricosene and cis-9-pentacosene, although no gain of female hydrocarbons was observed (***Fernandez et al., 2010***). Brain specific depletion of *desat1*, which encodes a desaturase enzyme involved in pheromone biosynthesis, was shown to alter pheromone production (***Bousquet et al., 2012***). We noted the presence of unconventional sites that potentially could be targeted by

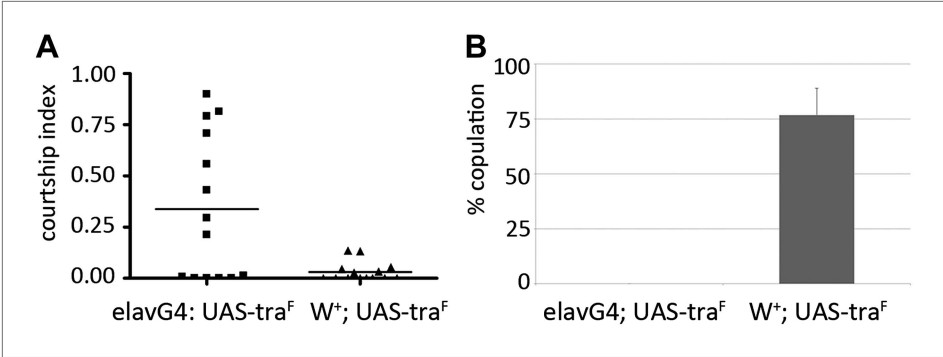

**Figure 8**. Transcript level of *dsx*[M], but not *dsx*[F], is affected by *miR-124* loss-of-function. Expression of *dsx*[M] (**A**) *dsx*[F] (**B**) transcripts measured by quantitative real-time PCR using RNA isolated from male flies of the indicated genotypes. *actin 42A* was used as an internal control for normalization. Data represent the average of four independent experiments ± SEM. **$p<0.05$, NS: not significant.

*miR-124* in the open reading frame and 5′ UTR of the *desat1* mRNA (**Figure 9**). The function of these sites has not been tested. If functional, *desat1* could be overexpressed in the *miR-124* mutant. While the consequences of elevated Desat1 expression are not known, the possibility exists that *miR-124* might act via multiple targets in the CNS to indirectly modulate pheromone production in peripheral tissues. In moths, the neuropeptide PBAN has been linked to control of pheromone production, suggesting a role for neuroendocrine control of sexual differentiation (**Jurenka and Rafaeli, 2011**). Our findings provide evidence that *miR-124* regulation of *transformer* may act in the context of neuroendocrine control of male pheromone production.

## Materials and methods

### Fly stocks and genetics

Flies were maintained on standard yeast-cornmeal-agar medium at 25°C, 60% relative humidity on a 12:12 light-dark cycle. Canton-S was used as the wild-type control. In all experiments, *miR-124* mutants were a transheterozygous combination of two independently generated alleles. The *miR-124*[Δ4] and *miR-124*[Δ177] targeted knockout alleles are described in (**Weng and Cohen, 2012**). The original knockout alleles contain a *mini-white* genetic marker flanked by LoxP sites. Because *mini-white* can affect behavior, the marker was excised from the original *miR-124*[Δ177,w+] and *miR-124*[Δ4,w+] alleles by crossing to Cre-expressing flies, as described (**Chen et al., 2011**). *mini-white*-excised derivatives of *miR-124*[Δ177] and *miR-124*[Δ4] were each backcrossed to Canton S for six generations prior to behavioral tests. The deficiency line uncovering the *miR-124* locus used in **Figure 1A** is Bloomington stock BL7837.

**A**
```
desat1    5'  C   U    CU   ACA  AU        C 3'
              UGG CAU  UCA   CG  UGCCUU
              ACC GUA  AGU   GC  ACGGAA
miR-124   3' GA             G    GC      U 5'
```
```
desat1    5'  C       GCGGA            C 3'
              UGGGUGU      CGUGUGCUUU
              ACCGUA       GCGCACGGAA
miR-124   3' GA      AGUG             U 5'
```

**B**
```
elo68a    5'  A    CAUAU                U 3'
              UGGC      AUUC CC  UGCCUU
              ACCG      UAAG GG  ACGGAA
miR-124   3' GA            U   CGC     U 5'
```
```
elo68a    5'  C   A     ACACG         C 3'
              UGG AUUCGCCG      GCUUU
              ACC UAAGUGGC      CGGAA
miR-124   3' GA   G       GCA        U 5'
```

**Figure 9**. *miR-124* sites on *desat1* and *elo68α* transcripts. Left: sequences of two potential *miR-124* sites in *desat1* transcript. Top: a 6-mer site in the coding sequence common to all the isoforms; Bottom: an unconventional site with 2 GU base pairs in the 5′UTR of *desat1-RC* isoform. Right: sequences of two potential *miR-124* sites on *elo68α* transcript. Seed pairing in both sites are weak. All of these sites are unconventional and it is uncertain whether they would show regulation by *miR-124*. Their function has not been tested experimentally.

For genetic rescue experiments, the *mini-white* reporter in *miR-124* $^{\Delta177}$ RMCE allele was replaced with a *miR-124* hairpin fragment, as described (**Weng and Cohen, 2012**). The rescued mutant flies were homozygous for this chromosome (**Figures 2, 4 and 6**). The *miR-124* promoter Ga4 transgene is described in (**Weng and Cohen, 2012**). The UAS-tra-RNAi transgene was Bloomington Stock #28,512.

## Behavior assays

For courtship assays, males were collected at late pupal stage and aged individually for 5 days; target flies were collected at late pupal stage and aged for 5 days in groups of 20/vial. Behavior assays were performed 2–4 hr before lights off, 25°C, 60% relative humidity under normal ambient light.

Courtship assays were carried out as described (**Demir and Dickson, 2005**). For male–female assays, Canton-S virgin females served as mating targets. 5-day-old socially naïve Canton-S, *miR-124* mutants or *miR-124*-rescue males were tested. Courtship behavior was videotaped for 45 min after a virgin female and a test male were introduced into the courtship chamber by gentle aspiration. The courtship index is the proportion of time males spend courting within a 10-min observation period.

## Female receptivity assay

Male–female courtship assays were carried out as described (**Demir and Dickson, 2005**). 5-day-old socially naïve Canton-S males were paired individually with either 5-day-old Canton-S or 5-day-old *miR-124* virgin females. Courtship behavior was videotaped for 45 min after pairing. The percentage of females that accepted copulation by CS males was recorded for each genotype.

Male–male courtship assays: on the day of the experiment, target males were briefly anesthetized on ice and decapitated with a razor blade before being introduced into courtship chambers. Individual intact test males were gently aspirated into the chamber containing a decapitated target and the behavior of the test males was recorded for 45 min.

## Female choice assays

Round chambers of 10 mm diameter and 4 mm height were used for the mating competition assay. Mutants and wild-type male flies were collected at late pupal stage and isolated in standard food vials. On the fourth day post eclosion, mutants and controls were anaesthetized briefly and marked with acrylic paint at the back of the thorax. On the fifth day, a mutant and a wild-type with different colors were introduced into a courtship chamber containing a Canton-S virgin female and were videotaped for 70 min. The percentage of copulation success for both mutants and controls was measured.

## Aggression assay

The fighting chamber was 16 mm in diameter and 9 mm in height. A food patch was introduced by pipetting 50 µl of melted standard fly food in the center of the chamber. Pairs of socially naïve 5 day-old male flies were aspirated gently into the fighting chamber. Behavior was recorded for 45 min. Experimental and control groups were videotaped simultaneously. Fighting latency measures the number of encounters until the first antagonistic encounter between the pair. Frequency reports the number of incidents, including lunging and fencing, observed in 30 min.

## Locomotion assay

5-day-old socially naïve CS or *miR-124* mutant males were individually aspirated into the courtship chamber used for the male–female courtship assay as described above. The activity of the fly was videotaped for 15 min by a Sony Camcorder and analyzed by ImageJ. The velocity of the fly in the first 10 min of observation was recorded.

## Cuticular hydrocarbon extraction

Flies were reared as for the behavior assays and aged in groups of 15–20 flies per vial. Six replicates of fifteen 5-day-old male flies were anaesthetized on ice and placed into 1.8 ml glass microvials with Teflon caps (s/n 224740; Wheaton, Millville, NJ). 120 µl of hexane (Fisher Chemicals, Pittsburgh, PA) containing 10 µg/ml of hexacosane (Sigma-Aldrich, St Louis, MO) standard was added into each vial and incubated at room temperature for 20 min. 100 µl of solvent was transferred into a new vial and evaporated under a gentle stream of nitrogen. Extract was stored at −20°C until analysis. At least three biological replicates were prepared per genotype.

## Gas chromatography–mass spectrometry (GCMS) analysis

Extracts were re-dissolved in 60 µl of hexane and transferred into GC-MS vials (Supelco). Analysis was run in a 5% phenyl-methylpolysiloxane (DB-5, 30 m length, 0.32 i.d., 0.25 µm film thickness, Agilent, Santa Clara, CA) column and GCMS QP2010 system (Shimadzu, Kyoto, Japan) with an initial column temperature of 50°C for 2 min and increment to 300°C at a rate of 15 °C/min in splitless mode. The relative signal intensity for each hydrocarbon species was calculated by dividing the area under the chromatography peak by the total area under all of the peaks. The values from 3–6 replicate measurements were averaged.

## Pheromone perfuming

For application of synthetic compounds to target flies, 9 µg of synthetic cVA (Cayman Chemical Company Ann Arbor, MI) was diluted in 200 µl of hexane and introduced into a 1.8-ml glass microvial. The hexane was evaporated under a gentle flow of nitrogen, leaving the compound as a residue coating the bottom of the vial. Flies were briefly anaesthetized on ice, transferred to coated vials in groups of seven, and subjected to three vortex pulses lasting 20 s each, with 10 s pauses between each pulse. The perfumed flies were allowed to recover for about 1 hr in fresh vials with standard food. Six flies from each group were used for behavioral tests and the remaining fly was subjected to hydrocarbon analysis by Direct Analysis in Real Time mass spectrometry to monitor effective transfer of the test compound to the flies.

## Analysis of perfumed insects using Direct Analysis in Real Time mass spectrometry (DART MS)

The atmospheric pressure ionization time-of-flight mass spectrometer (AccuTOF-DART, JEOL USA, Inc.) was equipped with a DART interface and operated in positive-ion mode at a resolving power of 6000 (FWHM definition). Mass accuracy is within ±15 ppm. The DART interface was operated using the following settings: the gas heater was set to 200°C, the glow discharge needle was set at 3.5 kV. Electrode 1 was set to +150 V and electrode 2 was set to +250 V. $He_2$ gas flow was set to 2.5 l/min. Under these conditions, mostly protonated ($[M + H]^+$) molecules are observed. Using clean forceps, an anaesthetized fly was picked up by both wings, making sure not to damage the fly. The fly was placed in a stream of charged helium gas until peaks of triacylglycerides start to appear. All fly samples were placed approximately in the same location in the DART source for the same amount of time in order to obtain reproducible spectra. Six flies from each genotype were measured. Polyethylene glycol (Sigma-Aldrich) was used as calibrant. Relative quantification of compound abundance was performed by normalizing the areas under the signal corresponding to cVA ($[M + H]^+$ 311.29) to the tricosene signal ($[M + H]^+$ 323.36). DART MS is unable to differentiate isoforms of tricosene therefore the tricosene signal represents the summed signal intensity from 5, 7, and 9-Tricosene. Tricosene was selected as the normalization peak due to the unaltered levels in mutants compared to CS controls in GC-MS.

## Statistics

Statistical analysis for behavior assays and hydrocarbon quantification was done using Prism 4 (GraphPad Software, La Jolla, CA). For behavior data, a nonparametric Mann–Whitney test was used to compare two samples. Kruskal–Wallis test followed by Dunn's post-test was used to compare multiple samples. For hydrocarbon analysis, multi-way ANOVA followed by Tukey HSD post-test was performed.

## Transfection and luciferase assays

S2 cells were transfected in 24-well plates with 250 ng of miRNA expression plasmid or empty vector, 25 ng of firefly luciferase reporter plasmid, and 25 ng of Renilla luciferase DNA as a transfection control. Transfections were performed in triplicate in at least three independent experiments. 60 hr after transfection, dual luciferase assays (Promega, Madison, WI) were performed according to manufacturer's instructions.

## Acknowledgements

We thank Bruce Baker, Stephen Goodwin and Barry Dickson for fly strains and reagents, Kah-Junn Tan for technical support and D Foronda, H Herranz, J Varghese, LC Foo, J-M Kugler and WC Ng for helpful discussion. RW held a Singapore Millennium Foundation Scholarship. SC lab was supported by IMCB. JYY was supported by the Singapore National Research Foundation.

## Additional information

### Funding

| Funder | Author |
| --- | --- |
| Institute of Molecular and Cell Biology | Ruifen Weng, Stephen M Cohen |
| National Research Foundation of Singapore | Jacqueline SR Chin, Joanne Y Yew |

The funders had no role in study design, data collection and interpretation, or the decision to submit the work for publication.

### Author contributions

RW, Conception and design, Acquisition of data, Analysis and interpretation of data, Drafting or revising the article; JSRC, Conception and design, Acquisition of data, Analysis and interpretation of data; JYY, SMC, Conception and design, Analysis and interpretation of data, Drafting or revising the article; NB, Conception and design, Acquisition of data

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
