## [Decision Letter]

Thank you for sending your work entitled “*miR-124* controls male reproductive success in *Drosophila*” for consideration at *eLife*. Your article has been favorably evaluated by a Senior editor and 3 reviewers, of whom, Mani Ramaswami, is a member of our Board of Reviewing Editors.

The Reviewing editor and the other reviewers discussed their comments before we reached this decision, and the Reviewing editor has assembled the following comments to help you prepare a revised submission.

This manuscript describes male-specific behavioral defects in *miR-124* mutants, nicely demonstrating that *miR-124* is required for the development/expression of male-specific courtship and aggressive behavior in *Drosophila*. These behavioral defects of *miR-124* mutants, as well as associated changes in production of the pheromone cVA, are linked to increased expression of transformer (*tra*), a splicing factor that regulates the male-specific splicing of its downstream targets, *dsx* and *fru*. Several lines of evidence show *tra* to be a target of *miR-124*, and reduction of *tra* is shown to suppress *miR-124* mutant phenotypes. As cVA is a cuticular hydrocarbon of males flies that mediates male–female attraction, as well as male–male repulsion, several of the behavioral defects of *miR-124* are proposed to be explained by the dysregulation of one downstream target, *tra*, and its role in the production of sex-specific pheromones. Together, this supports a model in which endogenous *miR-124* (and therefore potentially other miRNAs) act to suppress phenotypes that may result from the leaky regulation of tissue and sex-specific mRNA splicing.

In terms of novelty and interest, this is appropriate for publication in *eLife*. However, several additional experiments and substantive revisions are required to strengthen some key conclusions. The most important of which are essential to convincingly explain the links between altered *tra* expression, altered pheromone production, and behavior.

1) As *tra* function in sex determination is mediated through its effects on *fru* and *dsx*, and a major hypothesis here is that increased levels of *tra* would result in leaky regulation of sex-specific splicing, it would be valuable to assess how the expression of these transcripts downstream of *tra* is affected *miR-124* mutants. Leaky *tra*^*F*^ would suggest the male makes less *Fru*, less *Dsx*^*M*^, and some *Dsx*^*F*^.

2) It is necessary to establish the cell type in which *miR-124* repression of *tra* occurs, as relevant to the production of cVA. Is *miR-124* expressed in oenocytes? Is reduction of *tra* in *miR-124* oenocytes sufficient to suppress *miR-124* mutant phenotypes? Is reduction of *tra* in the nervous system sufficient (if so, then this should be explained), or not, to suppress *miR-124* mutant phenotypes?

3) An alternative hypothesis for *miR-124* behavioral phenotypes is that increased levels of *tra* in *miR-124* mutants results in altered development/morphology of sex-specific neural circuitry. An analysis of the anatomy of *fru* or *dsx* expressing neurons in brains of *miR-124* mutant and *miR-124* mutant flies with reduced *tra* levels will allow a deeper understanding of the link between elevated *tra* expression in *miR-124* mutants and associated behavioral phenotypes.

4) Throughout the text and figure legends the authors should be explicit about which genotypes have been used, particularly for *miR-124* mutant males. Figure 1, what is the deficiency used? Figure 1, what is the *miR-124* mutant male genotype? Complete genotypes are also needed in Figure 2, Figure 4, and Figure 6.

5) Figure 1, the CI for the control flies is unusually low: what are the reasons for this? Male–female courtship values seem to show an unacceptably large distribution. It is not clear from these data (Figure 1) whether or not *miR-124* affects male–female courtship.

6) The origins, details, and properties of the *miR-124* mutant, rescuing transgene, sponge and *miR-124-Gal4* should be provided in the Materials and methods.

---

## [Author Response]

*1) As* tra *function in sex determination is mediated through its effects on* fru *and* dsx*, and a major hypothesis here is that increased levels of* tra *would result in leaky regulation of sex-specific splicing, it would be valuable to assess how the expression of these transcripts downstream of* tra *is affected* miR-124 *mutants. Leaky* tra^F^
*would suggest the male makes less* Fru*, less* Dsx^M^*, and some* Dsx^F^*.*

Less *dsx*^*M*^ transcript was detected in head samples from *miR-124* adult males. However, we did not observe an increase of *dsx*^*F*^ transcript. This is somewhat unexpected. The conventional model is that *tra* should switch splicing between the *dsx*^*M*^ and *dsx*^*F*^ forms. It is possible that production of the two *Dsx* splice products does not vary linearly with *tra*^*F*^ levels. We speculate on a possible molecular explanation in the revised Discussion (with data provided in Figure 8).

We were unable to measure *fru*^*M*^ levels by qPCR. None of the 6 pairs of primers tested gave a specific PCR product (multiple products formed).

*2) It is necessary to establish the cell type in which* miR-124 *repression of* tra *occurs, as relevant to the production of cVA. Is* miR-124 *expressed in oenocytes? Is reduction of* tra *in* miR-124 *oenocytes sufficient to suppress* miR-124 *mutant phenotypes? Is reduction of* tra *in the nervous system sufficient (if so, then this should be explained), or not, to suppress* miR-124 *mutant phenotypes?*

*miR-124* is not expressed in oenocytes.

Removing oenocytes does not affect cVA production (3). It is unlikely that the miRNA is acting in oenocytes, so we have not tested the effects of reducing tra in oenocytes in the *miR-124* mutant.

Several lines of evidence indicate that *miR-124* acts in the nervous system:

(A) Reduction of *miR-124* activity in neurons is sufficient to reproduce the male–male courtship phenotype. *miR-124* was depleted in neurons by expressing the *miR-124* sponge using *elav-Gal4*. Data are shown in the new Figure 5.

(B) *elav-Gal4* driven expression of *Tra*^*F*^ increased male–male courtship and reduced completion of male–female courtship (Figure 10). Elevated *Tra*^*F*^ in the CNS is sufficient to reproduce the effects of the *miR-124* mutant.Author response image 1.

(C) Selective depletion of *tra* mRNA in the CNS offset the male–male courtship phenotype in the *miR-124* mutant background. The level of male–male courtship was comparable in the *elav-Gal4*>UAStra^RNAi^ control compared to the *elav-Gal4*>UAStra^RNAi^ in the *miR-124* mutant. Data are shown in the new Figure 6.

*3) An alternative hypothesis for* miR-124 *behavioral phenotypes is that increased levels of* tra *in* miR-124 *mutants results in altered development/morphology of sex-specific neural circuitry. An analysis of the anatomy of* fru *or* dsx *expressing neurons in brains of* miR-124 *mutant and* miR-124 *mutant flies with reduced* tra *levels will allow a deeper understanding of the link between elevated* tra *expression in* miR-124 *mutants and associated behavioral phenotypes.*

This hypothesis would lead to the expectation of a change in the behaviour of the *miR-124* mutant male. The data on courtship behaviour do not support this view.

First, there was no difference in the behaviour of the *miR-124* mutant male toward control CS males in the male courtship assay (Figure 2). Second, interaction with behaviorally inert (decapitated) mutant males elicited behavioral changes in wild-type males. These observations cannot be explained by changes in the neural circuitry in the mutant male brain. We suggest that the pheromone experiments provide a more likely explanation for the phenotypes.

We have looked at *Fru*^*M*^ expression in *miR-124* mutant and control brains (*fru*^*P1*^*-gal4*>CD8RFP and anti *Fru*^*M*^). We saw no obvious differences. For the reasons above, we do not think it would be fruitful to look in depth for subtle changes.

*4) Throughout the text and figure legends the authors should be explicit about which genotypes have been used, particularly for* miR-124 *mutant males.*
Figure 1*, what is the deficiency used?*
Figure 1*, what is the* miR-124 *mutant male genotype? Complete genotypes are also needed in*
Figure 2*,*
Figure 4*, and*
Figure 6*.*

We intended to provide full genotype information in the Materials and methods section, but we now see that there were a few details missing. The text has been revised to provide complete information, as requested.

*5)*
Figure 1*, the CI for the control flies is unusually low: what are the reasons for this? Male–female courtship values seem to show an unacceptably large distribution. It is not clear from these data (*Figure 1*) whether or not* miR-124 *affects male–female courtship.*

The point of this experiment was to ask whether the reason for the low mating success shown in Figure 1 was due reduced courtship activity by mutant males.

Figure 1 shows that mating success was reduced for the mutant males. Figure 1 explore the stage at which the mutant males fail. B and C exclude early stage effects. Figure 1 was designed to ask if the defect in 1A was due to reduced courtship activity by the mutant males. For this we used decapitated female targets, to remove female behavioural input. The mutants did not seem to show reduced CI compared to the CS control when tested with behaviorally inert females.

The reviewers are correct to note that the variance is large in this data set. This is true for both the control and the mutant. We think it may be helpful to present the data as a scatter plot to give a better feel for the variance (Figure 11).Author response image 2.

The data is borderline significant at p<0.05 for the difference between the median of 56 pairs of flies. (The original Figure 1 showed the data comparing the means of 4 sets of 14 pairs. Analysed that way the difference was borderline not significant.)

For these experiments the mutant was backcrossed for 6 generations into the CS control background. The two populations should be very similar, except at the *miR-124* locus. The flies used for the experiments in Figure 1 were from the same amplified populations. We do not have an explanation for the variance and low CI, but note that whatever causes this in the control should also do so in the mutant (whether the cause is genetic or environmental in origin, we have controlled the two populations as best we can).

That said, the conclusion we draw from this experiment is that there is no evidence for reduced courtship activity by the mutant male. We are not trying to use this data as evidence in support of a behavioural difference. If the editors prefer, we are willing to remove this experiment. The result is not essential to the logical flow of the manuscript.

*6) The origins, details, and properties of the* miR-124 *mutant, rescuing transgene, sponge and* miR-124-Gal*4 should be provided in the Materials and methods*.

Done.